# Factored Neural Machine Translation on Low Resource Languages in the COVID-19 crisis

**Saptarashmi Bandyopadhyay**
University of Maryland, College Park
College Park, MD 20742
sapta.band59@gmail.com

## Abstract

Factored Neural Machine Translation models have been developed for machine translation of COVID-19 related multilingual documents from English to five low resource languages, Lingala, Nigerian Fulfulde, Kurdish Kurmanji, Kinyarwanda, and Luganda, which are spoken in the impoverished and strive-torn regions of Africa and Middle-East Asia. The objective of the task is to ensure that COVID-19 related authentic information reaches the common people in their own language, primarily those in marginalized communities with limited linguistic resources, so that they can take appropriate measures to combat the pandemic without falling for the infodemic arising from the COVID-19 crisis. Two NMT systems have been developed for each of the five language pairs – one with the sequence-to-sequence NMT transformer model as the baseline and the other with factored NMT model in which lemma and POS tags are added to each word on the source English side. The motivation behind the factored NMT model is to address the paucity of linguistic resources by using the linguistic features which also helps in generalization. It has been observed that the factored NMT model outperforms the baseline model by a factor of around 10% in terms of the Unigram BLEU score.

## 1 Introduction

The 2019-2020 global pandemic due to COVID-19 has affected the lives of all people across the world. It has become very essential for all to know the right kind of precautions and steps that everyone should follow so that they do not fall prey to the virus. It is necessary that appropriate authentic information reaches all in their own language in this period of infodemic as well, so that people can fluently communicate and understand serious health concerns in their mother tongue. Authentic information sites like FDA (FDA) and CDC (CDC) have published multilingual COVID related information in their websites. A machine translation system (Way et al., 2020) for high resource language pairs like German-English, French-English, Spanish-English and Italian-English has also been developed and published to enable the whole world to fight the pandemic and the associated infodemic as well.

The present work reports on the development of factored neural machine translation systems in five different low resource language pairs English – Lingala, English - Nigerian Fulfulde, English – Kurdish (Kurmanji), English - Kinyarwanda and English-Luganda. All the target languages are low resource languages in terms of natural language processing resources but are spoken by a large number of people. Moreover, the countries where the native speakers of these languages reside are facing COVID-19 crisis at various level.

Lingala (Ngala) (Wikipedia, c) is a Bantu language spoken throughout the northwestern part of the Democratic Republic of the Congo and a large part of the Republic of the Congo, as well as to some degree in Angola and the Central African Republic. It has over 10 million speakers. The language follows Latin script in writing. English - Lingala Rule based MT System (at Google Summer of Code) has been developed by students interning at the Apertium organization in the Google Summer of Code 2019 but no BLEU score has been reported.

Luganda or Ganda (Wikipedia, d) is a morphologically rich and low-resource language from Uganda. Luganda is a Bantu language and has over 8.2 million native speakers. The language follows Latin script in writing. An English - Luganda SMT system () has been reported that when trained with morphological segmentation at the pre-processing stage produces BLEU score of 31.20 on the Old

Testament Bible corpus. However, it is not relevant to our research due to the different medical emergency context captured by the corpora.

Nigerian Fulfulde (Wikipedia, e) is one of the major language in Nigeria. It is spoken by about 15 million people. The language follows Latin script in writing.

Kinyarwanda (Wikipedia, a) is one of the official language of Rwanda and a dialect of the Rwanda-Rundi language spoken by at least 12 million people in Rwanda, Eastern Democratic Republic of Congo and adjacent parts of southern Uganda. The language follows Latin script in writing.

Kurdish (Kurmanji) (Wikipedia, b), also termed Northern Kurdish, is the northern dialect of the Kurdish languages, spoken predominantly in southeast Turkey, northwest and northeast Iran, northern Iraq, northern Syria and the Caucasus and Khorasan regions. It has over 15 million native speakers.

The following statistics, as on 30th June 2020 and shown in Table 1, are available from the Coronavirus resources website hosted by the Johns Hopkins University (JHU) regarding the effects of COVID-19 in the countries in which the above languages are spoken.

In the present work, the idea of using factored neural machine translation (FNMT) has been explored in the five machine translation systems. The following steps have been taken in developing the MT systems:

1. The initial parallel corpus in the Translation Memory (.tmx) format has been converted to sentence level parallel corpus using the resources provided by (Madlon-Kay).

2. Tokenization and truecasing have been done on both the source and the target sides using MOSES decoder (Hoang and Koehn, 2008).

3. English source side after tokenization and truecasing have been augmented to include factors like Lemma (using Snowball Stemmer (Porter, 2001)) and PoS tags (using NLTK Tagger). No factoring has been done on the low resource target language side.

4. Byte Pair Encoding (BPE) and vocabulary are jointly learnt on original and factored dataset with subword-nmt tool (Sennrich et al., 2016).

5. BPE is then applied on the training, development and testing datasets for original and factored datasets with the subword-nmt tool (Sennrich et al., 2016).

6. The vocab file is obtained from training and development datasets of source and target files for both original and factored datasets.

7. MT system is trained on original and factored datasets.

8. Testing is carried out on original and factored datasets.

9. The output target data is post-processed, i.e., detokenized and detruecased for both original and factored datasets.

10. Unigram BLEU score is calculated on the detokenized and detruecased target side for both original and factored datasets.

## 2 Related Works

Neural machine translation (NMT) systems are the current state-of-the-art systems as the translation accuracy of such systems is very high for languages with large amount of training corpora being available publicly. Current NMT Systems that deal with low resolution (LowRes) languages ((Guzmán et al., 2019); (ws-, 2018)) are based on unsupervised neural machine translation, semi-supervised neural machine translation, pretraining methods leveraging monolingual data and multilingual neural machine translation among others.

Meanwhile, research work on Factored NMT systems (Bandyopadhyay, 2019); (Koehn and Knowles, 2017); (García-Martínez et al., 2016); (Sennrich and Haddow, 2016) have evolved over the years. The factored NMT architecture has played a significant role in increasing the vocabulary coverage over standard NMT systems. The syntactic and semantic information from the language is useful to generalize the neural models being learnt from the parallel corpora. The number of unknown words also decreases in Factored NMT systems.

## 3 Dataset Development

The five target languages and language pairs with English as the source language are shown in Table 2. The parallel corpus have been developed by several academic (Carnegie Mellon University, Johns

| Language | Country | Affected | Deceased |
|---|---|---|---|
| Lingala | Congo | 7039 | 170 |
| Luganda | Uganda | 889 | 0 |
| Nigerian Fulfulde | Nigeria | 25694 | 590 |
| Kinywarwanda | Rwanda | 1025 | 2 |
| Kurdish (Kurmanji) | Turkey | 199906 | 5131 |

Table 1: COVID 19 statistics in the countries speaking the five low resource languages

Hopkins University) and industry (Amazon, Apple, Facebook, Google, Microsoft, Translated) partners with the Translators without Borders (TICO-Consortium) to prepare COVID-19 materials for a variety of the world's languages to be used by professional translators and for training state-of-the-art Machine Translation (MT) models.

| Target Language | Language Pair |
|---|---|
| Lingala | en-ln |
| Nigerian Fulfulde | en-fuv |
| Kurdish Kurmanji | en-ku |
| Kinyarwanda | en-rw |
| Luganda | en-lg |

Table 2: Five low resource language pairs

The number of sentences in the training, development and testing datasets for each language pair have been mentioned in Table 3.

| Training | Development | Testing |
|---|---|---|
| 2771 | 200 | 100 |

Table 3: Number of lines in the training, development and testing datasets

The following factors have been identified on the English side:

1. Stemmed word–The lemmas of the surface-level English words have been identified using the NLTK 3.4.1. implementation of the Snowball Stemmer (Porter, 2001).

2. PoS Tagging has been done using the pos_tag function in the NLTK 3.4.1 on the English side of the parallel corpus.

During the experiments with various NMT architectures, the training, the development and test files were initially tokenized and then shared Byte Pair Encoding (BPE) was learned on the tokenized training corpus to create a vocabulary of 300 tokens in each of the experiments and then BPE was implemented on the tokenized training, development and testing files. The subword-nmt tool was used for the purpose of implementing Byte Pair Encoding (Sennrich et al., 2016) on the datasets to solve the problem of unknown words in the vocabulary. Accordingly, the corpus size remained the same but the number of tokens increases.

## 4 Experiment and Results

FNMT experiments were conducted in OpenNMT (Klein et al., 2017) based on PyTorch 1.4 with the following parameters:

1. Dropout rate - 0.4

2. 4 layered transformer-based encoder and decoder

3. Batch size=50 and 512 hidden states

4. 8 heads for transformer self-attention

5. 2048 hidden transformer feed-forward units

6. 70000 training steps

7. Adam Optimizer

8. Noam Learning rate decay

9. 90 sequence length for factored data

10. 30 sequence length for original data

Experiments have been conducted with factored datasets (Lemma, PoS tag attached to the surface word by a '|' on English side with no factors on the target side) and non factored datasets. The evaluation scores for each experiment setup are reported in the Unigram BLEU.

It is observed in Table 4 that there is around 10% significant improvement in the Unigram BLEU score in the factored neural architecture with greedy inference, indicating a drastic improvement in the

translation quality of the test dataset. The best Unigram BLEU scores are observed with Lingala (ln) as the target language. Moderate BLEU scores are observed with Nigerian Fulfulde (fuv) and Kurdish Kurmanji (ku) as target languages while the translation quality for Kinyarwanda (rw) and Luganda (lg) target languages can be improved significantly.

| Language Pair | Factored | Original |
|---|---|---|
| en-ln | 22.7 | 19.0 |
| en-fuv | 14.9 | 13.6 |
| en-ku | 11.8 | 10.9 |
| en-rw | 7.5 | 5.8 |
| en-lg | 5.5 | 3.4 |

Table 4: Unigram BLEU scores on the *test datasets* for the target languages

The results indicate that a bigger training dataset is essential to improve the translation quality and only 2771 lines of training data is not sufficient for this purpose. Using additional synthetic data generated with backtranslation as proposed in (Przystupa and Abdul-Mageed, 2019) can be a possible way forward.

## 5 Conclusion

In the present work, the non-factored and factored NMT systems developed for the five low resource language pairs with target languages as Lingala, Nigerian Fulfulde, Kurdish Kurmanji, Kinyarwanda and Luganda have been evaluated on Unigram BLEU scores. Since the corpus size is very small, multiBLEU evaluation scores have not been considered. NMT in the reverse direction, i.e., with English as the target language has not been attempted as the motivation was to translate authentic COVID-19 related information from English to the five low-resource target languages. It is observed that for all English to target language translation, the factored NMT system performs better in terms of the Unigram BLEU score. Future works will be carried out including more parallel data and incorporating synthetic data with backtranslation in the respective language pairs and incorporating lemma and PoS tag information on the target language sides.

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
