# OpenReview forum: "Factored Neural Machine Translation on Low Resource Languages in the COVID-19 crisis"
_EMNLP/2020/Workshop/NLP-COVID — Submitted to NLP-COVID19-EMNLP_

### Official Review · AnonReviewer1 · 2020-09-13
**Important topic, but definitely needs more work**

**Rating:** 3
**Confidence:** 4

**Review:**

This work presents NMT models for COVID-19 related documents for five low resource languages in Africa and Mid-East Asia. The topic presented is definitely crucial, and the paper is easy to follow. I appreciate the author detailed all procedures to train and evaluate the models.

However, there are a few key issues that should be resolved before moving forward:

1. Data and Modeling
  * The translation model only takes <3k data for training. I doubt if any useful information could be learned with this limited data. The author may consider transfer learning and data augmentation to mitigate the scarcity of the data.
  * The vocabulary contains only 300 tokens. Is there any specific consideration for this?

2. Evaluation
  * The paper uses unigram BLEU for evaluation. However, unigrams only capture limited signal, and it is hard to make system comparison and justify that factored NMT is better with a rather limited metric. BLEU-4 seems to be a better option.
  * Given the test dataset has only 100 sentences, it is important to obtain confidence intervals to support the argument of one system is better than another.
  * The author mentioned `... to ensure that COVID-19 related *authentic information* reaches the common people in their own language...` However, it is unclear whether the information was translated factually correctly. In the long run, it is crucial to check the correctness of the translated sentences to make sure all critical information was translated correctly.

---

### Official Review · AnonReviewer3 · 2020-09-21
**Insufficient data resources for the proposed approach**

**Rating:** 3
**Confidence:** 3

**Review:**

This work reports on a series of experiments using neural machine translation systems for translating English into five low-resource languages (Lingala, Nigerian Fulfulde, Kurdish Kurmanji, Kinyarwanda, Luganda) focusing on information related to COVID-19. To do this, parallel corpora of approximately 3000 sentence pairs for each low-resource language are applied to two NMT models.

While this paper discusses an important topic, the datasets used to train the specified translation models from English into the low-resource languages are too small, and a larger amount of sequence pairs and/or a different approach (e.g., using domain adaptation techniques) to this task is required.

Comments:
1. There is a missing reference at the end of Section 1 (English-Luganda SMT system).
2. Some of the information in the Introduction could be moved to the Related Work section.
3. The BLEU metric mentioned in the paper requires a reference which is not provided.
4. The NLTK library used in the experiments requires a reference which is not provided.
5. In Section 2, is the phrase “low resolution” mistakenly used for “low resource”?

---

### Official Review · AnonReviewer4 · 2020-09-25
**Evaluation of baselines over MT dataset for low-resource languages**

**Rating:** 3
**Confidence:** 4

**Review:**

This article describes experiments on MT of covid-related documents into low-resource languages. The article is well motivated, but it would help to focus on the challenges and current approaches for low-resource MT, including unsupervised methods. Existing research streams are mentioned in a sentence, without providing enough context. For work on the unsupervised paradigm, see for instance Artetxe et al. (2019). For gaining space some parts of the motivation (such as Table 1) could be removed. Another section where more information would be desired is on the used dataset. There is no mention of the process to translate the data and its quality, or whether it has been used for MT in previous work.

With regards to the experiment, it is limited to applying an standard supervised solution exploring only the use of factoring. It would be interesting to explore more creative solutions and evaluation frameworks. For instance, they claim that the training data is insufficient, and they could test a learning curve to provide more clarity on this aspect. Monolingual data could also be used to test extensions to the model.

On the evaluation, it would be interesting to provide context on what the BLEU scores mean for a practical system, and provide at least some qualitative error analysis.

Minor comments:

- P1: "very essential": remove "very"
- P1: missing reference for English-Luganda SMT
- P2: "one of the major languageS"
- P2: "one of the official languageS"
- P2: "low resolution": low resource

An Effective Approach to Unsupervised Machine Translation
Mikel Artetxe, Gorka Labaka, Eneko Agirre
In ACL 2019